

**Temperature and UV light affect the activity of marine cell-**
**free enzymes**
Blair Thomson[1], Christopher David Hepburn[1], Miles Lamare[1], Federico Baltar[1,2]
[1]Department of Marine Science, University of Otago, New Zealand
[2]NIWA/University of Otago Research Centre for Oceanography, Dunedin, New Zealand
*Correspondence to*: F. Baltar (federico.baltar@otago.ac.nz)
**Abstract**
Microbial extracellular enzymatic activity (EEA) is the rate-limiting step in the degradation of organic matter in
the oceans. These extracellular enzymes exist in two forms, cell-bound which are attached to the microbial cell
wall, and cell-free which are completely free of the cell. Contrary to previous understanding, cell-free
extracellular enzymes make up a substantial proportion of the total marine EEA. Little is known about these
abundant cell-free enzymes, including what factors control their activity once they are away from their sites
(cells). Experiments were run to assess how cell-free enzymes (excluding microbes) respond to ultraviolet
radiation (UVR) and temperature manipulations, previously suggested as potential control factors for these
enzymes. The experiments were done with New Zealand coastal waters and the enzymes studied were alkaline
phosphatase [APase], β-glucosidase, [BGase], and leucine aminopeptidase, [LAPase]. Environmentally relevant
UVR (i.e., *in situ* UVR levels measured at our site) irradiances reduced cell-free enzyme activities up to 87%
when compared to controls, likely a consequence of photodegradation. This effect of UVR on cell-free enzymes
differed depending on the UVR fraction. Ambient levels of UV radiation (KJ) were shown to reduce the activity
of cell-free enzymes, for the first time. Elevated temperatures (15°C) increased the activity of cell-free enzymes
up to 53% when compared to controls (10°C), likely by enhancing the catalytic activity of the enzymes. Our
results suggest the importance of both UVR and temperature as control mechanisms for cell-free enzymes. Given
the projected warming ocean environment and the variable UVR light regime, it is possible there could be major
changes in the activity of cell-free EEA and their contribution to organic matter remineralization in the future.

**1 Introduction**
Heterotrophic microbes are ubiquitous in the marine environment, recycling most of the organic matter available
in the oceans. The discovery of the microbial loop made clear that heterotrophic microbes are one of the most
important nutrient vectors in marine food webs (Azam and Cho, 1987; Azam et al., 1983). According to the size-
reactivity model, microbes selectively prefer high molecular weight dissolved organic matter (HMWDOM) due
to its superior nutritional value (Amon and Benner, 1996; Benner and Amon, 2015). The main obstacle for use





of HMWDOM by microbes is that these compounds are generally too large to be transported across microbial
cell membranes. Enzymatic hydrolysis outside of the cell is required to break HMWDOM down to smaller size
fractions (<600 Daltons) before uptake can occur (Weiss et al., 1991). Thus, microbial extracellular enzymatic
activity (EEA) is the process that initiates the microbial loop (Arnosti, 2011; Hoppe et al., 2002), and is
recognised as the rate limiting step in the degradation of organic matter in the oceans (Hoppe, 1991). This key
role has led to extracellular enzymes being referred to as "gatekeepers of the carbon cycle" (Arnosti, 2011).
There are two forms of EEA; cell-bound, which are attached to the outside of the microbial cell wall or reside in
the periplasmic space, and; cell-free, which are completely free of the cell, suspended in the water column. Cell-
free enzymes can come from a variety of sources in the marine environment including the sloppy grazing
behaviour of protists (Bochdansky et al., 1995; Hoppe, 1991), microbial starvation (Chróst, 1991), the lysis of
cells by viruses (Kamer and Rassoulzadegan, 1995) and the direct release by microbes in response to the
detection of appropriate substrates (Alderkamp et al., 2007). Up until recently, research on extracellular enzymes
has been mostly on cell-bound enzymes, as they were considered to be the only abundant form (Hoppe, 1983;
Hoppe et al., 2002). This lead to a view that cell-bound extracellular enzymes were the only form of ecological
significance (Chróst and Rai, 1993; Rego et al., 1985). However, studies have now shown that the second form,
cell-free extracellular enzymes, can make up a substantial proportion of the total extracellular enzyme pool
(Allison et al., 2012; Baltar et al., 2013; Baltar et al., 2010; Baltar et al., 2016; Duhamel et al., 2010; Kamer and
Rassoulzadegan, 1995; Li et al., 1998). This has been a major conceptual shift for research in marine enzymatic
activity, generating new research questions about what controls cell-free enzymes in the marine environment and
how they function (Arnosti, 2011; Arnosti et al., 2014; Baltar et al., 2010; Baltar et al., 2016).
One of the many consequences of this discovery is that cell-free enzymes can be decoupled temporally and/or
spatially from the microbial community that produces them (Arnosti, 2011; Baltar et al., 2010; Baltar et al.,
2016), since cell-free enzymes have long residence times after they are released lasting up to several weeks
(Baltar et al., 2013; Steen and Arnosti, 2011). The activity of cell-free enzymes away from their sites (cells) can
condition macromolecular DOC and organic surfaces for subsequent microbial growth. This action at a distance
complicates discerning links between producing microbes and their enzymes expression, as cell-free enzymes
have the potential to contribute to the availability of nutrients at a great distance from the releasing cell (Arnosti,
2011; Baltar et al., 2010; Baltar et al., 2016). It has been suggested that the history of the water mass may be
more informative in understanding current cell-free enzyme activities than the *in situ* microbial community
present at the time of sampling (Arnosti, 2011; Baltar et al., 2013; Baltar et al., 2010; Baltar et al., 2016; Kamer
and Rassoulzadegan, 1995).
There is only a limited number published investigations into the dynamics of cell-free enzymes (Baltar et al.,
2013; Baltar et al., 2010; Baltar et al., 2016; Duhamel et al., 2010; Kamer and Rassoulzadegan, 1995; Kim et al.,
2007; Li et al., 1998; Steen and Arnosti, 2011). These papers provide good evidence of the importance of cell-
free enzymes in the marine environment, but the controls for cell-free enzymes (once separated from the
microbial cell) are poorly understood (Arnosti, 2011). Steen and Arnosti (2011) tested the effect of ultraviolet
radiation (UVR) on cell-free enzymes directly, finding that a reduction in cell-free enzyme activity only at
artificially high UVR doses (i.e., UV-B intensity 5–10 times higher than *in situ*), with natural illumination
showing no significant effects of photodegradation. One recent study by Baltar et al. (2016) in the Baltic Sea



revealed strong correlations between seasonal temperature change and the proportion of cell-free to total EEA,
suggesting seawater temperature and/or solar radiation as the most obvious abiotic mechanisms for the control of
cell-free enzymatic activity. However, that was a field study of coastal waters, which includes the whole
microbial community and many potential interactions and effects that can co-occur (e.g. production/consumption
of free enzymes by microbes, variation in substrate concentration, etc.). Thus, to better understand the factors
affecting marine free EEA we need to test the effect of environmental factors on free EEA under controlled
conditions.
Here we isolated the free extracellular enzymes from a coastal site and specifically studied the effects of
temperature and UVR on the activity of three cell-free extracellular enzyme groups; alkaline phosphatase
(APase), an enzyme used to acquire phosphorus from organic molecules; β-glucosidase (BGase), a glycolytic
enzyme that targets carbohydrates groups, and; leucine aminopeptidase (LAPase), an enzyme associated with the
degradation of proteins. UVR treatments were hypothesised to reduce the activity of cell-free enzymes when
compared to dark controls by photodegradation, with a 'high UVR dose' treatments (including the entire UV-B
spectrum (280 to 320 nm) were hypothesized to have a stronger degradative effect on cell-free enzymes than
'low UVR dose' treatments (which only include a fraction of the UV-B spectrum, 280 to 305 nm). This was
based on the reported effects of UV-B on microbes and their metabolic rates including the total EEA (Demers,
2001; Herndl et al., 1993; Müller-Niklas et al., 1995; Santos et al., 2012). Compared to ambient temperatures
(10°C), cell-free enzymes exposed to high temperatures (15°C) were hypothesised to be more active, and
*viceversa*, due to the general relationship between temperature and catalytic activity in enzymes (Daniel and
Danson, 2010, 2013). Experiments carried out here are the first to directly test temperature effects on cell-free
enzymes alone, and to directly test the effect of UVR on cell-free enzymes in the Southern Hemisphere and
under *in situ* measured environmental-relevant UV-irradiances.

**2 Materials and methods**
**2.1 Study site, sampling and experiments preparation**
The experiments were conducted at the University of Otago's Portobello Marine Laboratory, situated on the
Otago Harbour, Dunedin, New Zealand (45.8281° S, 170.6399° E). Otago Harbour is a tidal inlet which has an
area of 46 km$^2$, consisting of two basins and with extensive sediment flats (Grove and Probert, 1999; Heath,
1975). The laboratory is based on the outer Otago harbour, which has waters similar in composition to coastal
seawater, owing to the rapid residence times for its waters exchanging with the open sea (Grove and Probert,
1999; Rainer, 1981). Samples were taken from the second meter of the water column off the marine laboratory's
wharf that extends into a deep tidal channel. All sampling and laboratory equipment used was prior sterilised by
triplicate rinses of 18 MΩ·cm high purity water (Milli-Q$^{TM}$) water before and after soaking in 10% hydrochloric
acid for >6 hours and oven dried at 60°C.  To separate the cell-free extracellular enzymes from the total
extracellular enzyme pool and the microbial community, samples were gently triple filtered through low protein
binding 0.22µm Acrodisc filters following published methods (Baltar et al., 2010; Kim et al., 2007). 50 ml glass
vials were filled with the 0.22 µm-filtered seawater for use in experiments. Bacterial abundance was determined




after both experiments by preserving samples in glutaraldehyde and processing using SYBR Green nucleic acid
stain with a BD Accuri C6 flow cytometer (BD biosciences, USA). This was to ensure that no significant
bacterial growth occurred after filtering or during the incubation. Bacterial abundance was reduced to less than
1% of the pre-filtered total and remained so during the 36-hour incubations.

**2.2 UVR experiments**

To determine *in situ* UVR irradiance and environmentally appropriate treatments for experiments, the
attenuation of UVR was measured through the upper 2 m of the water column on site using a LI-COR
LI1800UW spectroradiometer (LI-COR biosciences, USA. The spectroradiometer was factory calibrated using
NIST traceable standards.  Once this was determined, artificial lighting was installed in a controlled temperature
room, set to the ambient seawater temperature (10°C). The lighting consisted of two FS20 UV-R lamps (General
Electric, Schenectady NY, USA) and a full spectrum Vita-Lite 72 (Duro-Test, Philadelphia, PA, USA) lamp,
suspended above the samples.  These lights were height adjusted to yield an irradiance of 3.03 W m$^{-2}$ s$^{-1}$ UVR,
approximating UVR irradiances measured in the field at 2 m depth (3.5 W m$^{-2}$ s$^{-1}$).  Schott WG and GG long
pass filters (15 cm X 15 cm) with nominal cutoffs (50% T) in the UVB (280 nm, 305 nm) were placed over the
filtered cell-free enzyme seawater samples contained in glass vials, with either a 'high dose' (<280nm, 3.03 W
m$^{-2}$ s$^{-1}$, 130.8 kJ) or a 'low dose' (<305nm, 0.42 W m$^{-2}$, 18.1 kJ) of UVR. All light was blocked except that
which passed directly through the long pass filters. Controls were kept without light by wrapping the glass vials
containing the filtered cell-free enzyme seawater samples in several layers of aluminium foil, and were placed in
the same controlled temperature room. Readings of enzyme activity rates were taken of three replicates of each
treatment at 12 and 36 hours. Temperature inside the vials was also monitored to ascertain that the samples were
constantly kept at the desired temperature.

**2.3 Temperature experiments**

For the temperature experiments we utilised a large graded heat block system (see Lamare et al. (2014) for
design specifications). This heat block allowed for up to 15 replicate samples to be exposed to constant
temperature treatments over time. The heat blocks were tested five times a day for three days in advance with
blank samples to ensure the heat blocks were calibrated accurately; the variation in temperature was within
0.5°C of the target temperatures (i.e., 5, 10, and 15°C) in all measurements. These temperatures were selected
because 5 to 15 °C is the annual range of temperature in the sampling site, and 10$^{\circ}$C was the *in situ* temperature
at the time of sampling (unpublished data). All treatments were kept in the dark by wrapping the glass vials
containing the filtered cell-free enzyme seawater samples in several layers of aluminium foil. Readings of
enzyme activity rates were taken of three replicates of each treatment were at 6, 12, 24 and 36 hours. When
incubating these samples, each was put into a separate incubator which was set to the treatment temperature so to
avoid confounding the temperature treatments.



**2.4 Extracellular enzymatic activities assays**


We used the method for assessing extracellular enzymatic activity rates based on the hydrolysis of fluorogenic
substrate analogues developed by Hoppe (1983). The fluorogenic substrates: 4-methylcoumarinyl-7-amide
(MCA)-L-leucine-7-amido-4-methylcoumarin, 4-methylumbelliferyl (MUF)-β-D-glucoside and MUF-phosphate
were used to assess the leucine aminopeptidase, β-glucosidase and alkaline phosphatase activities, respectively.
Substrate concentrations of 100μM were used for each enzyme based on pre-established kinetics, tested in the
lab. 96-well falcon microplates were filled with six replicates of each of the three fluorogenic substrates (10μl)
and seawater (290μl) to make up 300μl reactions. Plates were read in a Spectramax M2 spectrofluorometer
(Molecular Devices, USA), with excitation and emission wavelengths of 365 and 445nm, both before, and after
3 hour incubations. All incubations were performed in the dark and kept in incubators set to *in situ* seawater
temperatures. Six samples without substrate addition served as blanks in each plate to determine the background
fluorescence of the samples, which were used to correct the activity rates in the plate readings before and after
incubation.

**2.5 Statistical analyses**


In all analyses, parametric assumptions were first checked using the Shapiro-Wilk test for normality and the
Levene's test for equal variance. Where appropriate, data was Log-transformed to meet normality assumptions
prior to analysis. Both experiments use two-way ANOVAs with an interaction term, with post hoc Tukey HSD
tests run to assess the individual significant effects between treatments. All analyses were run in the R software
environment (R Development Core Team, Austria).

**3 Results and Discussion**


**3.1 UVR experiments revealed photodegradation of cell-free enzymatic activities at environmentally**


**relevant levels**
UVR overall significantly decreased cell-free APase when compared to dark controls ($p<0.001$, $F_{2,12}=15.85$, two-
way ANOVA) (Fig. 1a). Individual significant effects between treatments in APase were seen as a significant
decrease in activity in the low-dose treatment relative to the dark control at 12 h ($p<0.05$, Tukey HSD), and
between the dark control and both the high and low UV-dose treatment at the 36-hour sampling point ($p<0.05$,
Tukey HSD). BGase cell-free activity was not significantly affected by UVR ($p=0.53$, $F_{2,12}=0.67$, two-way
ANOVA). UVR had a significant overall effect on LAPase, decreasing the cell-free activity when compared to
dark controls ($p<0.01$, $F_{2,12}=40.994$, two-way ANOVA) (Fig. 1c). Individual significant effects were seen in
LAPase, showing after 12 h a significant decrease in activity between the low and high at 12 h ($p<0.01$, Tukey
HSD), and after 36 h a gradual decrease from high to low dose ($p<0.05$, Tukey HSD), and dark control to both
low and high dose ($p<0.001$, Tukey HSD).
These experiments revealed a significant reduction in cell-free extracellular enzymatic activity for both APase
and LAPase in response to UVR, consistent with the predicted photodegradation; which was not evident for





BGase. This was the first time that UVR has been demonstrated to reduce cell-free enzymatic activities at
environmentally relevant intensities. The only previous study (Steen and Arnosti, 2011) did show a reduction in
the cell-free extracellular enzymatic activity of APase and LAPase but only at artificially high UVR intensities
where UV-B was 5–10 times more intense from artificial lamps in the lab than outdoors.  Interestingly, they
could not show significant UVR effects on BGase at any treatment level, which is consistent with the present
study.
Both APase and LAPase showed the strongest effect of UVR at the 36-hour sampling point, suggesting a UV-B
dose-dependent response.  LAPase also showed a gradual decrease in the effect between the low and high UVR
treatments, which suggests the increase in UV-B irradiances also enhanced the degree of photodegradation. UV-
B has been demonstrated to be a highly active part of the spectrum for degrading DNA in general (Dahms and
Lee, 2010; Sinha and Häder, 2002), with specific effects of UV-B on total extracellular enzymatic activities
previously reported (Demers, 2001; Herndl et al., 1993; Müller-Niklas et al., 1995; Santos et al., 2012).
However, it is important to distinguish these previous studies from the cell-free enzyme experiments performed
here. Those previous studies tested the response of the entire microbial community, for total extracellular
enzymatic activity, based on the assumption that UVR affects the organism (source of enzymes) directly. What
is shown in this study is that UVR affects cell-free exclusively without the need to impact the source organism.
The effects of UVR were different among the enzymes assessed, which may be of importance as some enzymes
could be more impacted by UVR than others. For example, in this study, APase and LAPase were more affected
by UVR than BGase, which could change the spectrum of extracellular enzyme activity in the surface of the
ocean.  The resulting higher BGase relative to APase or LAPase, could potentially condition macromolecular
DOC composition by hydrolysing relatively less proteins than carbohydrates in response to UV.  In turn, it is
conceivable that any change in the enzyme spectrum due to variability in UVR light could cause a loss of
productivity (e.g. due to a decrease in the inorganic P made available through APase activities), as the nutrients
made available by extracellular enzymes may not be in suitable ratios for the effective growth of microbes
(Arnosti et al., 2014; Häder et al., 2007).
**3.2 Temperature experiments revealed enhanced catalytic activity of cell-free enzymes**
Temperature significantly increased cell-free APase at the high temperature of 15$^{o}$C when compared to the
ambient control of 10$^{o}$C (p<0.01, $F_{2,24}$=11.57, two-way ANOVA) (Fig. 2a). APase activity was significant
increased, after 6 h, in the high relative to the low temperature (p<0.001, Tukey HSD), after 12 h between low
and high temperature (p<0.001, Tukey HSD), and control and high treatments (p<0.05, Tukey HSD). Cell-free
BGase showed a similar pattern of increased activity in response to higher temperature but it was not significant
(Fig 2b). This lack of significant differences in cell-free BGase in response to temperature could be due to a
relatively high variability in EEA among the high temperature (15$^{o}$C) treatments. LAPase significantly decreased
in the low temperature treatment (5$^{o}$C), relative to the ambient control (p<0.01, $F_{2,24}$=13.97, two-way ANOVA)
(Fig 2c). LAPase cell-free activity significantly increased between the low and high temperature treatments at
the 6h and 12h time points (p<0.05, Tukey HSD). The temperature effect was dependent on time, finding
significant effects after 6 and 12h, but not later for any of the studied enzymes.





The relationship found between temperature and cell-free activity is consistent with the general pattern of
increased catalytic activity of enzymes in relation to temperature (Daniel and Danson 2013). The positive
relationship between temperature and the activity of cell-free enzymes observed in this study is contrary to the
negative relationship between temperature and the proportion of cell-free relative to total EEA measured in a
seasonal field study in the Baltic Sea (Baltar et al., 2016). However, it is important to take into consideration the
fact that the study by Baltar et al. (2016) took place over a much longer temporal scale (1.5 years) and included
the whole microbial community; whereas in this study different factors were teased apart by focusing only on the
cell-free enzymes. This is supported by Baltar et al. (2016) where the proportion of cell-free relative to total EEA
was significantly negatively correlated to prokaryotic heterotrophic production, suggesting that the low
temperature preserves the constitutive activity of the cell-free enzymes better (than warm temperature) due to a
reduction in the metabolism of heterotrophic microbes that would reduce the consumption/degradation of
dissolved enzymes. The exclusion of heterotrophic microbes from our samples precluded this effect (i.e.,
heterotrophic degradation/consumption of free enzymes) of temperature from occurring, and allowed us to tease
apart the effect directly on the cell-free enzymatic activities. This also highlights the importance of scales when
dealing with microbial oceanographic processes.
Moreover, the observed time dependence of the effect of temperature on cell-free enzymes (with effects
noticeable in short time scale of ≤12 h), together with the tendency for stronger UVR effect after 36 h than 12 h,
might suggest a potential different scale in the response of cell-free enzymatic activity to UVR and temperature,
where the catalytic effect of temperature occurs faster than the UVR photodegradation, but more research would
be required to confirm this hypothesis.

**Conclusions**
Overall, temperature and UVR were both demonstrated as potential control mechanisms for the activity of
marine cell-free enzymes, providing a baseline for future research. This is the first report revealing the effects of
photodegradation of cell-free enzymes at environmentally relevant levels of UVR, and the effects of enhanced
temperature on the catalytic activity of marine cell-free enzymes. Environmentally relevant UVR had a
significant photodegradative effect that might be enzyme-specific (affecting APase and LAPase but not BGase),
with the potential to alter not only the rates of cell-free EEA but also the spectrum of enzyme expression in the
seawater. Alteration of the cell-free EEA spectrum from UVR variability, could have ecological and
biogeochemical implications like the conditioning of macromolecular DOM (i.e., affecting DOM composition
by hydrolysing some DOM compounds more relative to others), and the change of the elemental ratio of some
nutrients (e.g., affecting the availability of inorganic P due to a change in APase activity), with implications for
productivity and nutrient cycling. Additionally, given the variable UVR light regime spatially and temporally
(i.e. the 150% increase in UV-B in polar regions during spring-time ozone depletion, Smith et al., 1992) and the
documented anthropogenic changes in ocean temperature (Chen et al., 2007), it is probable that the activity of
cell-free EEA and their contribution to organic matter remineralization might be affected in the future, if not
already.





**Acknowledgements**
We would like to thank the team of technicians out at Portobello Marine Laboratory, most notably, Linda
Groenewegen and Reuben Pooley. This research was supported by a University of Otago Research Grant to FB.
The authors declare that they have no conflict of interest.





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



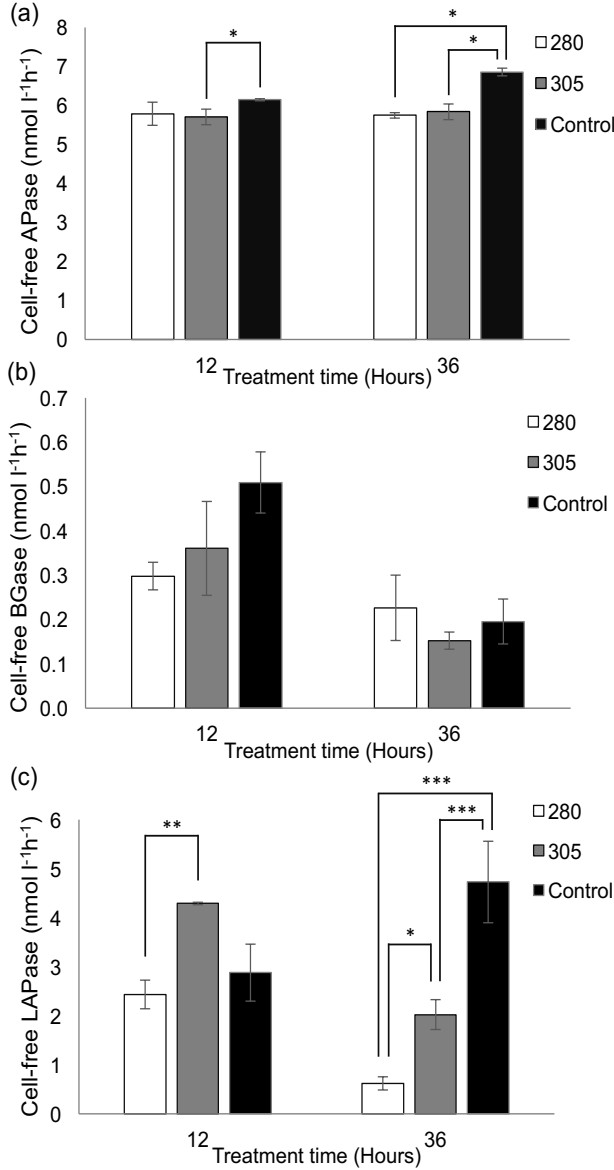

**Figure 1.** Results from UVR experiments showing the mean (±SE) cell-free extracellular enzyme activity for alkaline phosphatase (a), beta-glucosidase (b), and leucine aminopeptidase (c), under a high dose (of 280nm and above) and a low dose (of 305nm and above) in comparison to dark controls. Asterisks above graphs represent individual significant effects between treatments in post hoc Tukey test (*<0.05, **<0.01, ***<0.001) (N=3).



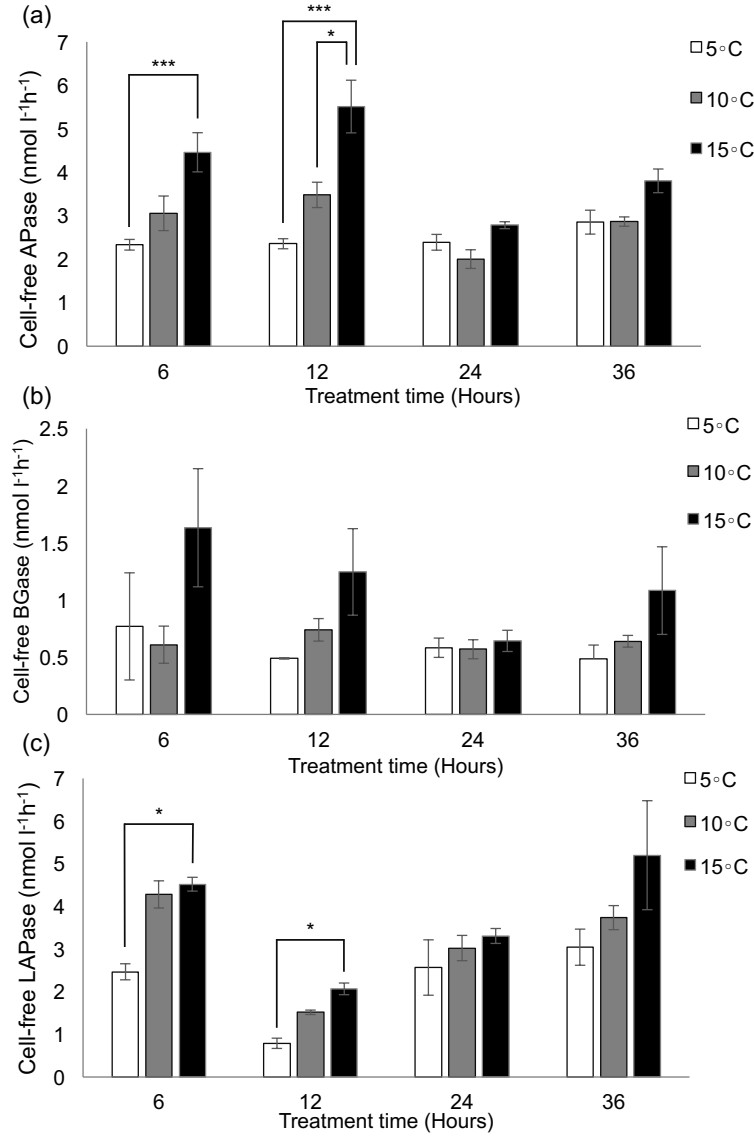

**Figure 2.** Results from temperature modification experiments showing the mean (±SE) cell-free extracellular enzyme activity for alkaline phosphatase (a), beta-glucosidase (b), and leucine aminopeptidase (c), under a high (15°C) and a low temperature (5°C) treatments in comparison to ambient controls (10°C). Asterisks above graphs represent individual significant effects between treatments in post hoc Tukey test (*<0.05, **<0.01, ***<0.001) (N=3).