# Peer review of "Temperature and UV light affect the activity of marine cell- free enzymes"

_Biogeosciences, 2017_

## Referee Comment (RC1) · Anonymous Referee #1 · 5 Mar 2017

General comments The manuscript deals with a subject of utmost importance. Hydrolytic enzymes are essential for microorganisms to process DOM, and in the climate change situation the alteration of the environmental conditions will modify the hydrolysis rates of polymers, and consequently the functioning of the carbon cycle. In this respect, the analysis of the effect of UVR and temperature on the free hydrolytic enzymes becomes relevant and even necessary. The manuscript shows for the first time the reduction of the activity of the free enzymes by the UVR at environmental intensities and this is a substantial contribution. However, regarding the effect of temperature the aim of the study is unclear, the design of the experiments is confusing and needs to be explained or improved, and the analysis and interpretation of the results also requires significant changes. General problems are the high variability of data, the variability of the controls in the UVR experiments, as well as the erratic pattern of the variation in

time of the measurements in the temperature experiments.

Specific comments 1. The aim of this work is to study "the effects of temperature and UVR on the activity of three cell-free extracellular enzyme groups" (L80-81). However, the measurements of activity were made in the dark and in situ temperature. Samples were exposed to different UVR doses and at different temperatures for 6- 36 h but the hydrolytic activity was measured in the dark and in situ temperature with 3 hours of incubation (L152-153). The authors should explain why the activity was not measured under the same conditions as the samples. This experimental approach allows detecting changes in the molecular state of the enzymes when they are exposed to different doses of UVR and different temperatures, but the fact that samples return for 3 h to in situ conditions makes it difficult to transfer the results to the ecosystem. Authors should explain the ecological sense of keeping the enzymes at 5°C or 15°C from 6 h to 36 h and then measuring the activity for 3 h at 10°C. I think that this is an important point that should be clarified to facilitate the understanding of the manuscript. 2. L104. In my opinion the treatment of the material used in the experiments should not be called sterilization. I think that the term decontamination would be more appropriate since sterilization destroys all living cells, included spores. 3. L107. The filtration process is critical because the filtration pressure can break the cells and release their contents, resulting in an enrichment of the filtrate. The authors claim that the filtration was gentle but, could the authors point out what they mean by gentle? What filtering pressure was used? 4. L148. In my experience the saturating concentrations are usually different in APase, BGase and LAPase, and frequently LAPase requires higher saturating concentrations than BGase and APase. The authors write that the concentration of substrate was established in previous kinetic experiments but should show some information about these experiments. Both environmental factors, UVR and temperature, can affect to the kinetic parameters and this should be taken into account. For example, changes in temperature do not only affect to the hydrolysis rate but also can modify the affinity of the enzymes and therefore the saturating concentration, 100 $\mu$M can be saturating at 10 °C, but no saturating at 5 °C. If molecules of enzymes are

affected by UVR and temperatures, the kinetic parameters (Vmax and Km) will also be affected. Some kinetic experiments with different UVR doses and temperatures would also significantly improve the manuscript because would show modifications in the enzyme molecules. 5. The introduction highlights the quantitative importance of the free enzymes and according to this it would be convenient to show, somewhere in the results, the percentage of total hydrolytic activity that dissolved enzymes represent in the analyzed samples. 6. It is not clear if figures correspond only to one UVR experiment (Fig 1) and one temperature experiment (Fig 2) or they show the average data of several experiments. If there are several experiments it would be more appropriate to show each experiment on one separate figure in order to reduce the standard errors. If data correspond to a single experiment it is not enough to reach any conclusion and the experiments must be repeated to find a common pattern. 7. Regarding the discussion, the exposure of free enzymes to different doses of UVR and different temperatures during 6-36 h also provides information on the stability of enzymes under different conditions, but it is not discussed in the manuscript although authors have experience on this issue. The authors detect differences between low and high dose of UVR and low and high temperature, but do not compare the evolution of the activity with time although there are important changes. 8. Fig 1. For both BGase and LAPase controls varied between 12 and 36 h and both showed high standard errors. The controls should keep stable for 36 h unless the stability of enzymes is affected. In the case of APase and LAPase the activity of the controls increased from 12 h to 36 h, while for BGase decreased. Were the differences statistically significant? If so, how do you explain these changes? In the case of BGase there was a general decrease from 12 to 36 h and the variability between replicates is so large that it possibly masks the effect of radiation, reason why there is not enough support to suggest that the effect of UVR is enzyme-specific and more experiments are required. 9. Fig 2. This figure tries to represent the effect of temperature on the activity of the free enzymes but it does reflect the stability of the enzymes to different temperatures over 36 h. In the case of BGase again the variability of data makes any comparison difficult. For LAPase the activity

decreased after 12 h but increased after 24 h and again after 36 h at the three temperatures. The effect of temperature could be expected to be maintained or increased over time. Authors should try to explain these erratic tendencies. 10. L230-234. I have some problems with this paragraph. The effect of UVR on APase is evident at 12 h and also for LAPase there are differences between doses at 12 h. Thus, the scales are not as different as the authors claim.

---

## Referee Comment (RC2) · Anonymous Referee #2 · 4 Apr 2017

This is an interesting paper, but a bit overly simplistic and seems to miss much of its potential. The fact that enzymes are affected by UV should not be surprising (they are complex organic molecules and the literature is replete with photochemistry). What are the structures of these enzymes? Since the result is different, whats different about the structures of the enzymes that suggests differences in sensitivity to UVR?

Nowhere do the authors address whether the effect is on the enzyme or perhaps the substrate? Whats the structure of the substrates, will they absorb UV?

The exposure methodology is unclear, the samples were placed in glass vials but were they irradiated through the glass (blocking much UV) or left open and irradiated from the top?

Spectrum of the lab light source is very different from the spectrum found in seawater,

lamps are a necessary evil, but a bit over simplification to say they had environmentally relevent irradiance. Why not do the incubations in situ in UV transparent containers (quartz, teflon, polyethylene?)

Finally, the discussion misses some classic literature - there were numerous papers published in the 80's from John Paul's lab on extracellular nucleases (DNAse)

---

## Author Comment (AC1) · 19 May 2017

Reviewer #1 General comments; The manuscript deals with a subject of utmost importance. Hydrolytic enzymes are essential for microorganisms to process DOM, and in the climate change situation the alteration of the environmental conditions will modify the hydrolysis rates of polymers, and consequently the functioning of the carbon cycle. In this respect, the analysis of the effect of UVR and temperature on the free hydrolytic enzymes becomes relevant and even necessary. The manuscript shows for the first time the reduction of the activity of the free enzymes by the UVR at environmental intensities and this is a substantial contribution. However, regarding the effect of temperature the aim of the study is unclear, the design of the experiments is confusing and needs to be explained or improved, and the analysis and interpretation of the results also requires significant changes. General problems are the high variability of data, the

variability of the controls in the UVR experiments, as well as the erratic pattern of the variation in time of the measurements in the temperature experiments

Author response to Reviewer #1 general comment: We thank the reviewer for the constructive comments on this manuscript. We have taken them on board and our responses to reviewer comments, including modifications to the manuscript, are detailed below. We hope the manuscript is clearer now and satisfies the reviewer.

REVIEWER COMMENT 1 by Referee #1: The aim of this work is to study "the effects of temperature and UVR on the activity of three cell-free extracellular enzyme groups" (L80-81). However, the measurements of activity were made in the dark and in situ temperature. Samples were exposed to different UVR doses and at different temperatures for 6- 36 h but the hydrolytic activity was measured in the dark and in situ temperature with 3 hours of incubation (L152-153). The authors should explain why the activity was not measured under the same conditions as the samples. This experimental approach allows detecting changes in the molecular state of the enzymes when they are exposed to different doses of UVR and different temperatures, but the fact that samples return for 3 h to in situ conditions makes it difficult to transfer the results to the ecosystem. Authors should explain the ecological sense of keeping the enzymes at 5âŮęC or 15âŮęC from 6 h to 36 h and then measuring the activity for 3 h at 10âŮęC. I think that this is an important point that should be clarified to facilitate the understanding of the manuscript.

Author response: In terms of the measurements for the temperature experiments; the enzyme assays were incubated at each of the three-respective treatment temperatures, so they were measured under the same conditions as the treatments. Thank you for pointing this confusion out; this has now been amended in the text (p. 6, l.169)(p. 6, l.185-186). The UVR experiments were all run at the in situ temp of 10 °C. When it came to the incubations for the enzyme assays, it was not clear the UVR dose would be consistent with the treatment level through the 96-well plates, so these were run in the dark, with the effect on EEA being the UVR treatment prior to the incubation.

Moreover, it is not appropriate to run the EEA assay under UVR since this radiation can affect the substrates analogues used in the assay. This has also been clarified (p. 5, l.154-155).

REVIEWER COMMENT 2 by Referee #1: 2. L104. In my opinion the treatment of the material used in the experiments should not be called sterilization. I think that the term decontamination would be more appropriate since sterilization destroys all living cells, included spores.

Author response: We agree with the reviewer, this has been modified accordingly (p. 4, l.123).

REVIEWER COMMENT 3 by Referee #1: L107. The filtration process is critical because the filtration pressure can break the cells and release their contents, resulting in an enrichment of the filtrate. The authors claim that the filtration was gentle but, could the authors point out what they mean by gentle? What filtering pressure was used?

Author response: We have removed the word gentle from the manuscript. Seawater was filtered using syringe filters as in previous works; we simply wanted to get across that the seawater was not forced through the filters in any way, but in hindsight, this is probably not necessary and was removed.

REVIEWER COMMENT 4 by Referee #1: L148. In my experience the saturating concentrations are usually different in APase, BGase and LAPase, and frequently LAPase requires higher saturating concentrations than BGase and APase. The authors write that the concentration of substrate was established in previous kinetic experiments but should show some information about these experiments. Both environmental factors, UVR and temperature, can affect to the kinetic parameters and this should be taken into account. For example, changes in temperature do not only affect to the hydrolysis rate but also can modify the affinity of the enzymes and therefore the saturating concentration, 100 $\mu$M can be saturating at 10 âŮęC, but no saturating at 5 âŮęC. If molecules of enzymes are affected by UVR and temperatures, the kinetic parameters (Vmax and Km) will also be affected. Some kinetic experiments with different UVR doses and temperatures would also significantly improve the manuscript because would show modifications in the enzyme molecules

Author response: We agree with the reviewer that the saturating concentrations can change, in fact in our preliminary saturation curves we did with water from the study site the saturating concentration was different for the different enzymes (i.e., around 83 $\mu$M, 57 $\mu$M and 39 $\mu$M for LAPase, than APase and BGase, respectively). We believed that in order to simplify confounding factors (because of all those different factors that the reviewer mention that can affect the saturating concentration) and with the aim to better compare the rates between the different enzymes, the best option was to use the same concentration for all the enzymes, which was saturating for all. Nevertheless, we have included a statement mentioning about the potential influence of temperature and/or UV on the saturating concentration of the EEA (p. 6, l.178-181).

REVIEWER COMMENT 5 by Referee #1: The introduction highlights the quantitative importance of the free enzymes and according to this it would be convenient to show, somewhere in the results, the percentage of total hydrolytic activity that dissolved enzymes represent in the analysed samples.

Author response: We have included a new Table (Table 1) including the information requested.

REVIEWER COMMENT 6 by Referee #1: It is not clear if figures correspond only to one UVR experiment (Fig 1) and one temperature experiment (Fig 2) or they show the average data of several experiments. If there are several experiments it would be more appropriate to show each experiment on one separate figure in order to reduce the standard errors. If data correspond to a single experiment it is not enough to reach any conclusion and the experiments must be repeated to find a common pattern.

Author response: The data of those plots responds to one of each form of experiment. But we believe that the experiments were carefully well replicated (3 biological + 6 technical replicates per sample/treatment), statistically supported and worth publishing; a first stepping-stone towards more complicated and sophisticated experiments in the near future.

REVIEWER COMMENT 7 by Referee #1: Regarding the discussion, the exposure of free enzymes to different doses of UVR and different temperatures during 6-36 h also provides information on the stability of enzymes under different conditions, but it is not discussed in the manuscript although authors have experience on this issue. The authors detect differences between low and high dose of UVR and low and high temperature, but do not compare the evolution of the activity with time although there are important changes Fig 1. For both BGase and LAPase controls varied between 12 and 36 h and both showed high standard errors. The controls should keep stable for 36 h unless the stability of enzymes is affected. In the case of APase and LAPase the activity of the controls increased from 12 h to 36 h, while for BGase decreased. Were the differences statistically significant? If so, how do you explain these changes? In the case of BGase there was a general decrease from 12 to 36 h and the variability between replicates is so large that it possibly masks the effect of radiation, reason why there is not enough support to suggest that the effect of UVR is enzyme-specific and more experiments are required.

Author response: The reviewer is right about the potential use of the experiment to learn more about the stability of the enzymes. However, the differences between the controls at 12h relative to 36h were not statistically significant for any of the enzymes. Nevertheless, we have included a sentence in the text specifying that the temporal differences in the controls were not statistically significant (p. 7, l.214-215).

REVIEWER COMMENT 8 by Referee #1: Fig 2. This figure tries to represent the effect of temperature on the activity of the free enzymes but it does reflect the stability of the enzymes to different temperatures over 36 h. In the case of BGase again the variability of data makes any comparison difficult. For LAPase the activity decreased after 12 h but increased after 24 h and again after 36 h at the three temperatures. The effect

of temperature could be expected to be maintained or increased over time. Authors should try to explain these erratic tendencies.

Author response: The higher variability of BGase was only found in the 6h time and in the 15°C, but not so much in the others. Moreover, the 10°C control of BGase was remarkably stable during all the length of the experiment (and showing low variability). So we believe that the data is good enough to allow for comparison. The more dynamic/erratic pattern observed for LAPase might be related to potential changes in adsorption/desorption and binding/unbinding of proteins/amino acids. Nevertheless, we believe that the fact of always having a control at every time point accounts for potential changes in this and other confounding factors.

REVIEWER COMMENT 9 by Referee #1: L230-234. I have some problems with this paragraph. The effect of UVR on APase is evident at 12 h and also for LAPase there are differences between doses at 12 h. Thus, the scales are not as different as the authors claim.

Author response: We have deleted this paragraph.

---

## Author Comment (AC2) · 19 May 2017

Author response to Reviewer #2 We thank the reviewer for the constructive comments on this manuscript. We have taken them on board and our responses to reviewer comments, including modifications to the manuscript, are detailed in the following:

Reviewer #2 REVIEWER COMMENT 1 by Referee #2: This is an interesting paper, but a bit overly simplistic and seems to miss much of its potential. The fact that enzymes are affected by UV should not be surprising (they are complex organic molecules and the literature is replete with photochemistry). What are the structures of these enzymes? Since the result is different, what's different about the structures of the enzymes that suggests differences in sensitivity to UVR?

Author response: Although we agree with the reviewer that the effect of UVR on free

enzymes could be expected, it had not been shown how marine produced cell-free enzymes were affected by UVR. We had a hypothesis based in fundamental theory and we applied that hypothesis to the marine environment, within a context where cell-free enzymes happen to be very important, and in an environment that happen to frequently fluctuate in temperature. It is difficult to tell at this point what the differences could be due to exactly in terms of the structure of the enzymes. For that we would need to perform more sophisticated protein structural research which is far from our scope here. The reality is that the majority of marine cell-free enzymes are poorly characterized and understood. There is likely to be structural differences between the glycolytic and proteinous enzymes for example which could affect their relative sensitivity to UVR, but a claim such as this would be a postulation/speculation at this point.

REVIEWER COMMENT 2 by Referee #2: Nowhere do the authors address whether the effect is on the enzyme or perhaps the substrate? What's the structure of the substrates, will they absorb UV?

Author response: We are not sure whether the reviewer refers to the natural substrates in the seawater sample or the artificial substrates used in the EEA assay. In the first instance, it is a good point that UV could affect the substrate (e.g. proteins, carbohydrates, etc), so we have now included this possibility in the discussion (p. 7, l.216-217). . For the latter, it is not a problem as the incubated plates themselves (which were supplemented with the artificial substrates) were not exposed to UVR, thus the artificial substrates were not exposed to UVR directly.

REVIEWER COMMENT 3 by Referee #2: The exposure methodology is unclear, the samples were placed in glass vials but were they irradiated through the glass (blocking much UV) or left open and irradiated from the top?

Author response: They were left open and irradiated from the top. This has been clarified in the methods (p. 5, l.150-151).

REVIEWER COMMENT 4 by Referee #2: Spectrum of the lab light source is very different from the spectrum found in seawater, lamps are a necessary evil, but a bit over simplification to say they had environmentally relevant irradiance. Why not do the incubations in situ in UV transparent containers (quartz, teflon, polyethylene?)

Author response: We agree with the reviewer that in-situ experiments would be much closer to reality in terms of a UVR dose. Our aim was to have the greater number of conditions/factors as controlled as possible to avoid other confounding factors. These experiments are part of a series which will eventually test multi-stress patterns, including both UVR and temperature; temperature being much harder to control in situ. We had already specified in the abstract that by "environmentally relevant irradiance" we mean that the authors tested and then used a dose level measured in-situ (p. 1, l.20-21).

REVIEWER COMMENT 5 by Referee #2: Finally, the discussion misses some classic literature - there were numerous papers published in the 80's from John Paul's lab on extracellular nucleases (DNAse)

Author response: These papers from John Paul's lab have now been reviewed. Thank you for pointing these out; some of these references were added to the discussion (p. 8, l.231-233).

END OF REVISION